# The Role of Service Providers' Resilience in Buffering the Negative Impact of Customer Incivility on Service Recovery Performance

**Valentina Sommovigo** [1,2,*], **Ilaria Setti** [1] 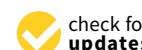 **and Piergiorgio Argentero** [1]

[1]   Department of Brain and Behavioral Science, Unit of Applied Psychology, University of Pavia,
     Piazza Botta, 11, 27100 Pavia, Italy; ilaria.setti@unipv.it (I.S.); piergiorgio.argentero@unipv.it (P.A.)
[2]   Department of Personnel & Employment Relations, Kemmy Business School, University of Limerick,
     Limerick V94 T9PX, Ireland
*   Correspondence: valentina.sommovigo@universitadipavia01.it

**Abstract:** In the service sector, customer-related social stressors may weaken employees' well-being, impairing job-related outcomes. Drawing on the Conservation of Resources theory and on the psychology of sustainability, fostering personal resources become critical to encourage service providers who can effectively manage such job demands. This study investigated how customer-related social stressors and customer orientation influence service recovery performance and whether resilience buffers the negative effects of customer incivility on service recovery performance. One hundred and fifty-seven Italian customer-contact employees completed a questionnaire analyzing customer incivility, customer-related social stressors, resilience, customer orientation, and service recovery performance. Regression analyses and SEMs were conducted. Although all customer-related social stressors indirectly and negatively influenced service recovery performance by increasing burnout symptoms, customer incivility only exerted a direct and detrimental impact on service recovery performance. Customer orientation was directly and positively associated with service recovery performance. Highly resilient employees were less affected by variations in service recovery performance across customer incivility levels. Within the psychology of sustainability framework, promoting resilient workplaces is crucial to foster healthy and sustainable work settings. Service organizations can greatly benefit from providing their employees with psychological resilience training programs, cultivating high customer-oriented attitudes through mentoring sessions, and hiring highly customer-oriented and resilient employees for customer-contact occupations.

**Keywords:** customer-related social stressors; resilience; customer orientation; service recovery performance; psychology of sustainability

## 1. Introduction

Nowadays, service organizations strive to deliver exceptional quality to their customers to succeed within increasingly competitive market environments [1]. Because of the high "people factor" [2], failures are often an inevitable part of the service delivery process. In this scenario, service recovery performance (SRP) plays a crucial role in recovering customers' loyalty and satisfaction [3], especially among Western countries where the service sector represents the main employment area, accounting for more than 60% of global Gross Domestic Product (GDP) [4]. For instance, in Italy such sector contributed around 70% to the employment rates and around 66% to the GDP in 2017 [5]. As a result, a large proportion of employees are frequently confronted with customer-related social stressors (CSSs) [6] which may produce detrimental effects on their well-being and SRP, depending

on individual characteristics. Indeed, some psychological characteristics may predispose workers to perceive customer encounters more or less favorably, influencing their reactions and vulnerability to CSSs [7,8]. In this context, resilience and customer orientation (CO) represent interesting variables, since they can be enhanced through specific trainings [9,10]. To date, the majority of research has focused on customer perceptions [11], whereas only a few studies have analyzed this topic from the service provider perspective [12]. To date, only a few studies have examined the direct association between CO and SRP, with previous investigations on this topic predominantly concentrating on organizational-level CO [13–15]. Moreover, to the best of our knowledge, no previous studies have analyzed resilience as a possible buffer for negative effects of customer incivility (CI) on SRP. To fill these gaps, drawing on the Conservation of Resource theory (COR) [16] and adopting the service provider perspective, the current study investigated how CSSs and individual-level CO influence SRP and whether resilience buffer the detrimental influence of CI on employees' SRP. Furthermore, this is one of the first studies to analyze this topic within the psychology of sustainability framework [17–19] which represents a promising research area for promoting healthy organizations [18] and improving employees' quality of life [20], all factors that are conducive to successful business [21,22] (From a psychological perspective, the word "sustainability" refers not only to balance current objectives with future aims without jeopardizing the latter by avoiding harmful actions within the ecological and socio-economic environment [23], but also to promote individual well-being by stimulating their enrichment, growth, and flexible change and by facilitating the acquisition of resources [17]). The rest of this paper is organized as following: (1) the next section briefly reviews the related literature around the relationships between CCSs and SRP as well as the protective role of resilience and CO in maintaining SRP, and then develops direct, mediating, and moderating hypotheses. (2) The second section describes the empirical setting of this study, including materials and methods. (3) The third section presents the statistical analyses and reports the empirical results, and (4) The final section discusses findings, limitations and practical implications, and then concludes the study.

## 1.1. Service Recovery Performance

SRP refers to "frontline service employees' perceptions of their own abilities and actions to resolve a service failure to the satisfaction of the customer" [24]. Correctly addressing customer discontent can lead to a host of positive outcomes, including reinforced positive word-of-mouth advertising, increased repurchase intentions and eventually customer patronage [25–29]. This may generate benefits for the whole organization in terms of profitability afterwards by fostering long-term seller-customer relationships [30] and by decreasing customer acquisition expenses [31]. Additionally, service providers can learn from recovery services and improve their performance—in terms of recovery speed and recovery quality, accordingly [11]. Moreover, evidence has shown that a good service recovery may not only compensate previous negative service experiences, but also increase post-failure customer satisfaction and loyalty, perceptions of relationship quality, and favorable company image beyond levels held before the service error (i.e., a phenomenon known as "service recovery paradox") [32–36]. This calls for acknowledging the importance of recovery encounters as "critical moments of truth" [37] because customers tend to view frontline service employees, who occupy "boundary spanning" roles [38], as organizational representatives and base their recovery evaluations mainly on the performance of these workers [39]. To date, there are various calls for more research regarding the factors stimulating employees' SRP [40,41]. By providing empirical evidence for the positive influence of CO and resilience on SRP, our study's analysis of SRP makes a meaningful contribution.

## 1.2. The Detrimental Impact of Customer-Related Social Stressors on Service Recovery Performance

To analyze potentially stressful customer-related events, we considered the four customer-related social stressor facets identified by Dormann and Zapf [6], as follows: (1) disproportionate customer expectations (DCE; i.e., customers' behaviors challenging what is considered reasonable from workers'

perspectives); (2) ambiguous customer expectations (ACE; i.e., unclear customers' requests); (3) disliked customers (DC; i.e., aversions customer-contact employees have to unpleasant customers who cause interruptions); (4) customer verbal aggression (CVA; i.e., verbal abuse perpetrated by a customer, with the clear intent to hurt a worker through offensive verbal expressions).

Although previous literature has mainly focused on CSSs in terms of such four dimensions, we decided to include also CI as a further form of CSSs. CI refers to low-intensity deviant, discourteous, and rude behaviors, perpetrated by a customer, with the ambiguous intent to harm an employee, in violation of workplace norms of mutual respect [42,43]. It differs from DCE, ACE, and DC because of the customer's ambiguous intent to harm an employee and it may include both verbal expressions (e.g., derogatory remarks) and disrespectful gestures (e.g., snapping fingers to get attention). CI can be conceptualized as a milder form of verbal aggression since it violates social norms through gestures or verbal expressions which lack the anger that characterizes verbal abuse [44–48]. Additionally, we hypothesized that Dormann and Zapf's CSSs could be placed on a continuum from DC to CVA with the addition of CI which could be included between customer expectations and CVA. In doing so, the aim of the current study was to extend the original model through the inclusion of CI. Evidence has been provided to support that even short-term negative customer encounters may undermine cognitive aspects relevant for customer service work and, therefore, result in lower quality performance [49]. Thereby, we expected that CSSs would directly impact on SRP, hypothesizing the following:

**Hypothesis 1 (H1).** *CSSs (H1a: DCE; H1b: ACE; H1c DC; H1d: CVA; H1e: CI) will be directly and negatively associated with SRP.*

*1.3. The Mediating Role of Burnout*

Previous investigations showed that CSSs [6] were positively related to, result in, and/or heighten emotional exhaustion (i.e., lack of energy and emotional fatigue) and cynicism (i.e., detachment from work and uncaring attitude towards customers) [50–62] which, therefore, might produce undesirable job outcomes, including impaired job performance [63–65]. A possible explanation could be drawn on the COR theory [16,66]. Indeed, according to this framework, service providers who experience high level of CSSs may perceive a threat to their working conditions and personal resources [16] or receive insufficient return of supplementary resources following significant resource investment (i.e., energy, time). Moreover, when employees are exposed to CSSs, they are likely to spend further resources to regulate negative emotions and think about their condition (e.g., by worrying about how they could avoid the situation). Whether service providers continue to be affected with resource loss without effectively compensating through resource replacement (i.e., by employing other resources to offset the loss), they may feel their resources are no longer sufficient to meet job demands and, therefore, be at increased risk of developing burnout. Therefore, once employees' emotional resources have been depleted, they may lack the energy to maintain their work efforts and successfully perform job-related tasks [67,68], such as SRP. Additionally, emotionally drained employees may try to reduce the loss of emotional resources by detaching themselves from customers (e.g., treating them as impersonal objects) to conserve their scarce resources [69]. To date, only a moderate amount of empirical research has examined the associations between CSSs, staff burnout, and SRP [6,70,71]; with a paucity of studies analyzing these variables in a single framework and supporting the role of burnout symptoms in mediating the impact of different CSSs on SRP [57,72]. To fill this gap, one of the main purposes of the current study was to investigate whether different CSSs would lead employees to experience burnout which, in turn, would decrease their ability to provide good SRP. Therefore, we formulated the following hypothesis:

**Hypothesis 2 (H2).** *Burnout symptoms will mediate the relationship between CSSs (H2a: DCE; H2b: ACE; H2c DC; H2d: CVA; H2e: CI) and SRP.*

### 1.4. The Importance of Personal Resources

In line with the psychology of sustainability and positive psychology frameworks, the enhancement of personal resources can play a crucial role in protecting employees from potential threats to their well-being and in maintaining their psycho-physical health over the life course [73]. This is consistent with the COR theory [16,66] which states that personal characteristics represent "resources to the extent that they generally aid stress resistance" [16] (p.517), suggesting that certain characteristics can be treated as personal coping resources. The differences in levels of stress-aiding personal characteristics may influence how individuals react to stress or loss of resources, making some individuals better at minimizing their losses and handling stressors. Moreover, personal resources produce other resources through resource gain spirals and engender resource caravans [74], such that the availability of greater resources protects against the risk of loss and enables to invest resources for the acquisition of further resources [75]. Conversely, poorly resourced individuals are more vulnerable to further resource "loss spirals" because they tend to be unable to offset additional losses and protect their remaining resources [16,66]. Among analyzed service providers' personal characteristics, we decided to focus on CO and resilience because the first has been identified as a protective factor against detrimental effects due to customer mistreatment [76–78], whereas the role of resilience in facing critical events has been widely recognized in the workplace violence literature [79].

### 1.5. The Positive Influence of Customer Orientation

CO is defined as an "employee's tendency or predisposition to meet customer needs in an on-the-job context" [80] (p. 111). CO refers to workers' beliefs about their capacity to satisfy customers' needs and desires by providing a courteous service. It captures the extent to which interacting with consumers is intrinsically pleasurable [10,81,82]. CO contributes to determine a service organization's business success [32,83–85] by decreasing negative individual and organizational-level consequences, such as burnout and turnover intentions [80,86–88], and by enhancing numerous job-related outcomes, including organizational commitment, job satisfaction [89], organizational citizenship behaviors [81], work engagement, job performance [80,86], and in particular SRP [13,90,91]. Assuming the COR theory perspective [66], CO can be considered as a personal coping resource [16,78] which makes customer-contact employees better at minimizing their losses because it predisposes them to seek additional resources to solve customers' problems [92] and cope better with CSSs [78]. Indeed, CO provides workers with an enduring reservoir of emotional and cognitive resources to pursue SRP [93]. Thereby, customer-oriented employees are predisposed to have a cooperative attitude, interpret their work environment through a customer service lens and display customer-satisfying behaviors [94]. Additionally, given their tendency to naturally read customers' needs and be emotionally stable when engaging in customer encounters [81], they tend to promptly respond to customer's problems with solutions and be highly motivated to be helpful towards clients [80]. Previous studies revealed that CO at the organizational level exerted a significant positive impact on service providers' SRP [13,15]. However, to date, to the best of our knowledge, the only study investigating the direct influence of individual-level CO on SRP was conducted by Choi and colleagues [56], obtaining the same result. This suggests that further empirical investigation on the direct relationship between individual-level CO and SRP is required. Taken together these findings, we expected that service providers' CO would be directly and positively associated with their SRP. Thus, we hypothesized the following:

**Hypothesis 3 (H3).** *CO will directly and positively influence SRP.*

### 1.6. Resilience as a Moderator

Resilience is defined as a state-like ability to rebound or bounce back from adversities, to deal with ongoing life challenges and succeed in maintaining equilibrium and positively adapting in the aftermath of stressful experiences [95–99]. Researchers consistently found that resilient people were likely to experience greater well-being [100], higher job performance [101–103], decreased distress

levels [104], better quality of life and lower severity of depression and anxiety [105,106]. Moreover, previous studies revealed that resilience (conceptualized as a dimension of psychological capital) had a positive influence on frontline employees' SRP [41,78,107]. In line with COR theory [66], resilience can be considered as a crucial psychological resource which can help people in facing professional challenges successfully, in addition to fulfilling positive resource gain spirals [108]. Indeed, resilient employees, who have a vast reservoir of personal resources, are better able to be flexible to stressful encounters and proactively prepare themselves for challenging demands. As a result, this psychological coping resource may allow workers to overcome obstacles and engage in goal striving. Moreover, resilience may shield workers from negative reactions to difficult situations by utilizing their resources and by replenishing them in an effective way [109]. Thereby, resilient workers are likely to respond effectively to numerous complaints and engage in extra-role behaviors to satisfy customer expectations. In other words, resilient employees could perceive customer encounters as challenging—rather than threatening—conditions and, thus, they would be more likely to maintain high SRP levels even in the face of uncivil clients. Therefore, this form of personal resource could moderate the detrimental effects of CI on SRP. By the same token, low resilience could intensify this harmful impact. Thus, we predicted the following:

**Hypothesis 4 (H4).** *Resilience will buffer the negative effects of CI on SRP, such that resilient employees will maintain high SRP even in the presence of high CI levels.*

As a conceptual framework, Figure 1 illustrates our proposed model, incorporating our hypothesized relationships. We expected that DCE, ACE, DC, CVA, and CI would directly (H1a, H1b, H1c, H1d, and H1e, respectively) and indirectly (through burnout symptoms; H2a, H2b, H2c, H2d, and H2e, respectively) negatively influence SRP. Moreover, we expected that CO would be directly and positively associated with SRP (H3). Additionally, we hypothesized that resilience would moderate the relationship between CI and SRP (H4).

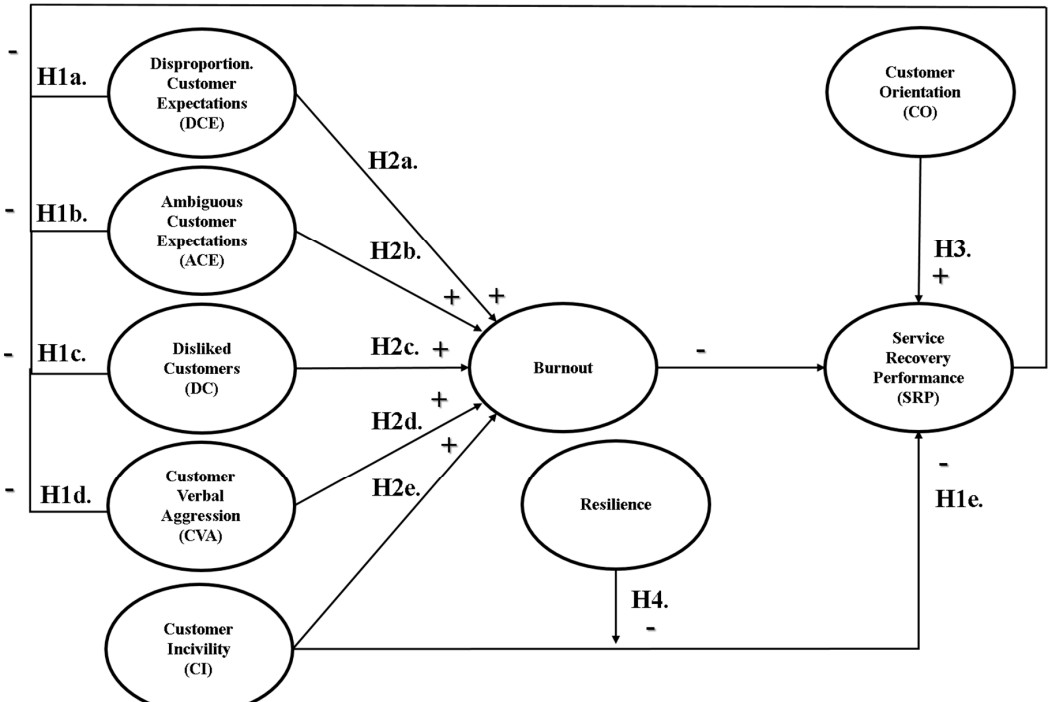

**Figure 1.** Proposed model regarding the relationships between customer-related social stressors (CSSs) and service recovery performance (SRP) as well as the protective role of resilience and customer orientation (CO) in maintaining service recovery performance (SRP).

## 2. Materials and Methods

### 2.1. Sample and Procedure

Participants were psychology students who were recruited using academic newsletter and e-mail system or were enrolled in psychology courses at University of Pavia. To participate, students were required to be working in a retail sale (e.g., shop assistant, cashier) or restaurant service (e.g., waiter, bartender) job for at least 6 months, be 18 years of age or older, and have at least a moderate amount of contact with the public, so that certain stressful events were likely to occur. Working students received extra course credit for taking part in the research. Once they voluntarily agreed to participate, we obtained informed consent from them and ensured them the anonymity and confidentiality of their responses. Then, they were invited to complete questionnaires which were administrated by professional trainees in Psychology within a laboratory setting. In total, 157 Italian students employed in customer-contact jobs took part to the present study. We excluded 3 participants who did not meet the study criteria (e.g., not employed in customer services) and 2 participants because they did not complete at least the 60% of the survey. The majority of respondents were female (78.30%) who spent on average the 85.91% of their working time in direct contact with the public. They were mainly employed as waiters or shop assistants. The average age was 25.27 years (SD = 5.59) with an average job tenure of 4.53 years (SD = 4.99) and an average job tenure in the current position of 2.95 years (SD = 4.99).

### 2.2. Measurements

Different individual characteristics were measured including resilience, CO and dispositional affectivity. Furthermore, participants were invited to answer questions concerning CI, CSSs, burnout, and SRP. All the variables were measured using scales taken or adapted from previously validated and published instruments. To assess resilience, trait affectivity, CSSs, and burnout, we used the Italian validated version of such scales. Because the other scales were originally written in English, they were subjected to a back and forward translation process. Firstly, the original English items were translated in Italian. Then, the forward translation was reviewed by a bilingual (in English and Italian) expert panel. Items that were suspected to be particularly sensitive to translation issues across cultures were translated back to English by an independent translator. The resulting version of the questionnaire was administrated to 30 pre-test respondents who were students employed in customer-contact occupations and, therefore, representative of research participants. They were systematically debriefed by asking them—for each item—what they thought was the meaning of a certain item, whether they could re-word that item using their own words, what sprang to their mind when they heard a specific expression, how they selected their answer. Finally, the modified version of the survey was discussed through two focus groups conducted by an experienced psychologist.

**Resilience** was assessed through 10 items from the Italian version [110] of the Connor–Davidson Resilience Scale [111]. Respondents reported how much they agreed with each statement concerning ways of dealing with problems and reacting to stressful situations (e.g., I can achieve goals despite obstacles) on a five-point Likert scale (0 = almost always false, 4 = almost always true). The reliability of this scale was 0.79.

**Customer orientation** was operationalized via 13 items from the Customer Orientation Scale [81]. Participants indicated to what degree they agreed with some statements concerning behavioral tendencies directed to meet customers' needs and expectations (e.g., I take pleasure in making every customer feel like he/she is the only customer), using a seven-point Likert scale (1 = strongly agree, 7 = strongly disagree). Cronbach's alpha for this scale was good ($\alpha = 0.89$). This scale has been broadly utilized by previous psychological studies, showing a satisfactory internal consistency (e.g., [56,77,78]).

**Trait Affectivity.** Positive affectivity (PA) and negative affectivity (NA) were evaluated using the Italian version [112] of the Positive and Negative Affect Scale (PANAS, [113]) which includes 10 positive and 10 negative mood states (e.g., concentrating for PA, upset for NA). This measure has

been validated in Italian samples (see [113]). Research subjects were asked to indicate how frequently they felt each of the listed emotional states in their workplace over the last two weeks on a five-point Likert scale (from 0 = very slightly or not at all to 5 = extremely). In line with previous studies conducted within the service sector, we decided to use trait affectivity (i.e., PA and NA) as a control variable [114,115]. The internal consistency of these scales was good ($\alpha$ = 0.89; $\alpha$ = 0.77, respectively).

**Customer incivility** was evaluated using 10 items from the Incivility from Customer Scale [48]. Participants indicated how frequently, in the last two weeks, had encountered rude customers in their actual workplace (e.g., Customers blamed you for a problem you did not cause) on a seven-point Likert scale (ranging from 1 = never to 7 = more than three times per day). This measure was developed based on a sample of working students who met criteria for research recruitment very similar to those applied to select our participants (i.e., working in retail sale or restaurant service occupations for at least 6 months). In addition, this scale has been widely used by previous investigations on workplace incivility, showing good reliability (e.g., [116]). In the current study, the reliability of the scale was very good ($\alpha$ = 0.92).

**Customer-related social stressors** were assessed through 22 items from the Italian version [117] of the Customer-related social stressors scale (CSSs scale, [6]) which analyzed to what extent participants believed some statements concerning encounters with customers were true in relation to their work experience, with response choices on a seven-point Likert scale (ranging from 1 = not at all true to 7 = absolutely true), in line with Dormann and Zapf's study [6]. Specifically, disproportionate customer expectations were evaluated via eight items (e.g., Some customers always demand special treatment; $\alpha$ = 0.90). Ambiguous customer expectations (e.g., One wishes are often contradictory; $\alpha$ = 0.82) and disliked customers (e.g., One has to work with hostile customers; $\alpha$ = 0.77) each included four items. Customer verbal aggression was measured through five items (e.g., Customer often shout at me; $\alpha$ = 0.76). We utilized this scale because our study was theoretically drawn on Dormann and Zapf's [6] classification that we decided to adopt since it reflects general categories suitable for different work environments and covers a broad range of customer behaviors. Additionally, this measure was developed based on a sample of three service sectors' employees (i.e., travel agency employees, shoe store sales clerks, and flight attendants) and, therefore, it was applicable for our sample. This scale has been widely used by previous studies on CSSs, showing a high degree of internal consistency among the construct items (e.g., [56,78,118]). In the current study, the reliability of the whole scale was 0.92.

**Burnout** was measured using ten items from the Italian version [119] of the Maslach Burnout Inventory [120], including two sub-scales: emotional exhaustion (5 items, e.g., "I feel emotionally drained from my work") and depersonalization symptoms (5 items; "I have become less enthusiastic about my work"). Consistent with Cordes and colleagues' contention [121] that decreased personal accomplishment represents a consequence of burnout rather than a distinct symptom of the condition, and according to Bakker and co-workers [63], we did not include such dimension in our conceptual framework; instead we concentrated on the impact on customer-contact employees' SRP as a result of burnout considered in its dimensions of emotional exhaustion and cynicism. All items were scored on a seven-point frequency Likert scale (ranging from 0 = never to 6 = daily). The internal consistency of the scale was good ($\alpha$ = 0.88).

**Service recovery performance** was evaluated using five items [13] (e.g., Considering all the things I do, I handle dissatisfied customers quite well) that investigated to what degree respondents agreed with some statements regarding the perceptions of being able to manage customer complaints and recover from service failures. The responses were obtained on a seven-point Likert scale (from 1 = strongly disagree to 7 = strongly agree). This scale, which has the advantage of its short length (5 items), has been widely applied by previous psychological studies which assessed SRP within service organizations (e.g., [56,59,77,78]). The reliability of the scale was 0.75.

## 3. Results

### 3.1. Statistical Analyses

The data were first explored for descriptive statistics and correlations using SPSS 20 statistical program for Windows [122]. Then, all CSSs were added to a regression model to simultaneously predict SRP. To test the mediating role of burnout in the relationship between CSSs and SRP, we conducted mediation models using Mplus Version 7 [123]. Fit models were examined using the root mean squared error of approximation (RMSEA, [124]; values of 0.05 are taken as good fit, 0.05–0.08 as moderate fit, [125]), the standardized root mean square residual (SRMR; a value less than 0.08 is considered a good fit, [125]), the comparative fit index (CFI, [126]; values between 0.90 and 0.95 indicate acceptable fit, [125]) and the Tucker–Lewis index (TLI, values between 0.90 and 0.95 indicate acceptable fit, [127]). To test direct effects of personal characteristics (i.e., CO, resilience, PA and NA) on SRP in a combined model, all individual features were added to a regression model having SRP as dependent variable. The regression analyses were conducted using enter variable selection in which variables were randomly selected and entered since we did not have research evidence to hypothesize a certain order. Subsequently, to examine the moderating effect of resilience on the relationship between CI and SRP, a moderation model was carried out using Mplus Version 7 [123], while controlling for CO, PA and NA. The goodness of the model was evaluated by comparing it in terms of BIC (Bayesian Information Criterion) and AIC (Akaike Information Criterion) comparative indices with three competing models. Lower values of AIC and BIC indicate a better fit and the model with the lowest AIC and BIC is the best fitting model.

### 3.2. Hypotheses Testing

Firstly, data were explored by conducting descriptive statistics and correlations among the study variables. As shown by Table 1, all CSSs were significantly and positively correlated with each other, except for DC. Similarly, all CSSs were significantly and positively associated with burnout, except for DC. This suggests that working students who were more frequently exposed to ACE were more likely to experience also DCE, CI and CVA. Additionally, employees who were affected from ACE ($r = 0.37$, $p < 0.01$), DCE ($r = 0.37$, $p < 0.01$), CI ($r = 0.38$, $p < 0.01$) and CVA ($r = 0.20$, $p < 0.05$) might be at increased risk of developing burnout symptoms. Although SRP was negatively associated with all CSSs, the correlations with CI ($r = -0.29$, $p < 0.01$) and DCE ($r = -0.17$, $p < 0.05$) only were significant. This means that service providers who were confronted with CI and DCE were more likely to react by reducing their SRP. Moreover, SRP was positively related to burnout ($r = -0.32$, $p < 0.01$), such that increased burnout symptoms resulted in decreased SRP. Regarding individual characteristics, SRP was positively associated with resilience ($r = 0.31$, $p < 0.01$), CO ($r = 0.55$, $p < 0.01$) and PA ($r = 0.30$, $p < 0.01$), whereas it was negatively related to NA ($r = -0.19$, $p < 0.05$). Additionally, burnout was positively correlated with NA ($r = 0.34$, $p < 0.05$) and negatively associated with CO ($r = -0.26$, $p < 0.01$); PA ($r = -0.50$, $p < 0.05$) and resilience ($r = -0.14$, ns), although this latter personal characteristic showed a non-significant correlation with SRP. Taken together these results, it seems that service providers high in resilience, CO and PA are more likely to maintain high SRP and psychological well-being levels, even when exposed to CSSs, differently from those high in NA. Then, regression analyses were carried out through enter variable selection using SPSS version19 [122] (see Table 2). All CSSs were added to a regression model to simultaneously predict SRP perceptions ($F_{(4,27)} = 4.27$, $p < 0.001$, $R^2 = 0.12$). CI was the only significant predictor of SRP, so that when employees reported to be exposed to high CI levels, they were more likely to perceive a decrease in their SRP ($\beta = -0.44$, $p < 0.01$). Thus, Hypothesis 1e was confirmed, whereas Hypothesis 1a, 1b, 1c and 1d were rejected.

**Table 1.** Descriptive, internal consistency and intercorrelations for study variables among service providers (N = 157).

| Measure | M | SD | 1 | 2 | 3 | 4 | 5 | 6 | 7 | 8 | 9 | 10 | 11 |
|---------|---|----|----|----|----|----|----|----|----|----|----|----|----|
| 1. CI | 2.27 | 0.98 | **0.89** | | | | | | | | | | |
| 2. CVA | 1.69 | 0.94 | 0.68 ** | **0.76** | | | | | | | | | |
| 3. DCE | 4.14 | 1.46 | 0.54 ** | 0.50 ** | **0.90** | | | | | | | | |
| 4. ACE | 3.85 | 1.38 | 0.40 ** | 0.41 ** | 0.67 ** | **0.82** | | | | | | | |
| 5. DC | 4.16 | 1.49 | −0.03 | −0.03 | 0.08 | −0.01 | **0.77** | | | | | | |
| 6. SRP | 4.79 | 1.02 | −0.29 ** | −0.11 | −0.17 * | −0.13 | −0.08 | **0.75** | | | | | |
| 7. Resilience | 2.83 | 0.51 | 0.06 | 0.09 | 0.13 | 0.04 | 0.08 | 0.31 ** | **0.77** | | | | |
| 8. CO | 5.51 | 0.68 | −0.24 ** | −0.170 * | −0.09 | 0.04 | 0.05 | 0.55 ** | 0.26 ** | **0.80** | | | |
| 9. PA | 3.49 | 0.74 | −0.20 ** | −0.160* | −0.14 | −0.17 * | 0.03 | 0.30 ** | 0.33 ** | 0.43 ** | **0.89** | | |
| 10. NA | 1.71 | 0.63 | 0.21 ** | 0.10 | 0.08 | 0.29 ** | −0.05 | −0.19 * | −0.35 ** | −0.04 | −0.12 | **0.87** | |
| 11. Burnout | 2.01 | 1.18 | 0.38 ** | 0.20 * | 0.37 ** | 0.37 ** | 0.10 | −0.32 ** | −0.14 | −0.26 ** | −0.5 * | 0.34 * | **0.88** |

Note. Boldfaced numbers on the diagonal represent Cronbach's alpha; M = means; SD = standard deviation; * $p < 0.05$; ** $p < 0.01$. CI = customer orientation; CVA = customer verbal aggression; DCA = disproportionate customer expectations; ACE = ambiguous customer expectations; DC = disliked customers; SRP = service recovery performance; CO = customer orientation; PA = positive affectivity; NA = negative affectivity.

To test whether burnout could mediate the associations between CSSs and SRP, mediation models were conducted using Mplus Version 7 [123]. As shown by Table 3, Hypotheses 2a, 2b, 2c, 2d, and 2e were supported. Indeed, all CSSs were negatively associated with burnout symptoms which, in turn, led employees to experience reduced SRP.

**Table 2.** Effects of customer-related social stressors and customer incivility on service recovery performance in a combined model.

| Variable | B | S.E. | β | t |
|----------|---|------|---|---|
| Customer incivility | −0.46 | 0.11 | −0.44 *** | −3.95 |
| Customer verbal aggression | 0.20 | 0.12 | 0.19 | 1.73 |
| Disproportionate customer expectations | −0.04 | 0.09 | −0.05 | −0.43 |
| Ambiguous customer expectations | −0.03 | 0.07 | −0.04 | −0.36 |
| Disliked customers | 0.07 | 0.06 | 0.11 | 1.20 |

Note. *** $p < 0.001$.

**Table 3.** Fit indices and standardized direct and indirect effects for mediation models analyzing the impact of each customer-related social stressors on service recovery performance via burnout.

| Model | $\chi^2$ | df | p | RMSEA | SRMR | CFI | TLI |
|-------|----------|----|----|-------|------|-----|-----|
| Model 1 | 446.34 | 264 | 0.000 | 0.06 | 0.07 | 0.90 | 0.90 |
| Model 2 | 229.07 | 140 | 0.000 | 0.06 | 0.07 | 0.93 | 0.92 |
| Model 3 | 341.165 | 218 | 0.000 | 0.06 | 0.07 | 0.93 | 0.92 |
| Model 4 | 225.256 | 140 | 0.000 | 0.06 | 0.07 | 0.94 | 0.92 |
| Model 5 | 221.09 | 138 | 0.000 | 0.06 | 0.07 | 0.94 | 0.93 |

| Standardized direct and indirect effects | | | |
|------------------------------------------|--|--|--|
| *Effects-Model 1* | Estimate | S.E. | p |
| **CI→Burnout→SRP** | **−0.11** | **0.05** | **0.024** |
| **CI→SRP** | **−0.31** | **0.10** | **0.002** |
| *Effects-Model 2* | Estimate | S.E. | p |
| **CVA→Burnout→SRP** | **−0.13** | **0.05** | **0.007** |
| CVA→SRP | −0.04 | 0.10 | 0.686 |
| *Effects-Model 3* | Estimate | S.E. | p |
| **DCE→Burnout→SRP** | **−0.17** | **0.05** | **0.001** |
| DCE→SRP | −0.07 | 0.10 | 0.463 |
| *Effects-Model 4* | Estimate | S.E. | p |
| **ACE→Burnout→SRP** | **−0.19** | **0.06** | **0.001** |
| ACE→SRP | −0.02 | 0.11 | 0.845 |
| *Effects-Model 5* | Estimate | S.E. | p |
| **DC→Burnout→SRP** | **−0.17** | **0.07** | **0.010** |
| DC→SRP | 0.03 | 0.30 | 0.764 |

Note. Boldfaced numbers indicate statistically significant models. df = degree of freedom; RMSEA = Root Mean Square Error of Approximation; SRMR = Standardized Root Mean Square Residuals; CFI= Comparative Fit Index; TLI = Tucker–Lewis Index; CI = customer orientation; CVA = customer verbal aggression; DCA = disproportionate customer expectations; ACE = ambiguous customer expectations; DC = disliked customers; SRP= service recovery performance.

To verify Hypothesis 3, a regression analysis was conducted to examine the impact of CO on SRP while controlling for resilience, PA and NA ($F_{(4,152)}$ = 20.42, $p < 0.001$, R = 0.59; see Table 4). As expected, CO directly and positively influenced employees' SRP perceptions ($\beta$ = 0.50, $p < 0.001$), whereas neither resilience nor affectivity traits were directly and significantly associated with SRP. CO was the only significant predictor of SRP, in a direction in line with what expected.

**Table 4.** Effects of customer orientation (CO) on SRP while controlling for resilience and affectivity traits.

| Variable | B | S.E. | $\beta$ | t |
|---|---|---|---|---|
| Customer orientation | 0.75 | 0.11 | 0.50 *** | 6.82 |
| Resilience | 0.26 | 0.15 | 0.13 | 1.77 |
| Positive affectivity | 0.04 | 0.10 | 0.03 | 0.36 |
| Negative affectivity | −10.20 | 0.11 | −10.12 | −1.74 |

Note. *** $p < 0.001$.

In order to test whether resilience could moderate the relationship between CI and SRP, a moderation model was conducted using Mplus Version 7 [123] while controlling for CO, PA, and NA. Hypothesis 4 was supported: resilience buffered the relationship between CI and SRP (see Figure 2 and Table 5), as indicated by the significant interaction term ($\beta$ = −0.08, $p < 0.05$) which had a negative sign and, then, indicated that resilience could exacerbated the negative effect of CI on SPR. More specifically, the conditional effects showed that working students who reported low ($\beta$ = −0.23, $p < 0.01$) or moderate ($\beta$ = −0.14, $p < 0.01$) resilience levels were at higher risk of experiencing impaired SRP as a result of CI when compared with those who were higher in this dimension. Examination of the interaction plot (see Figure 3) showed that high-resilient service providers tended to report approximately the same level of SRP regardless of the extent to which they were targeted of incivility from customers.

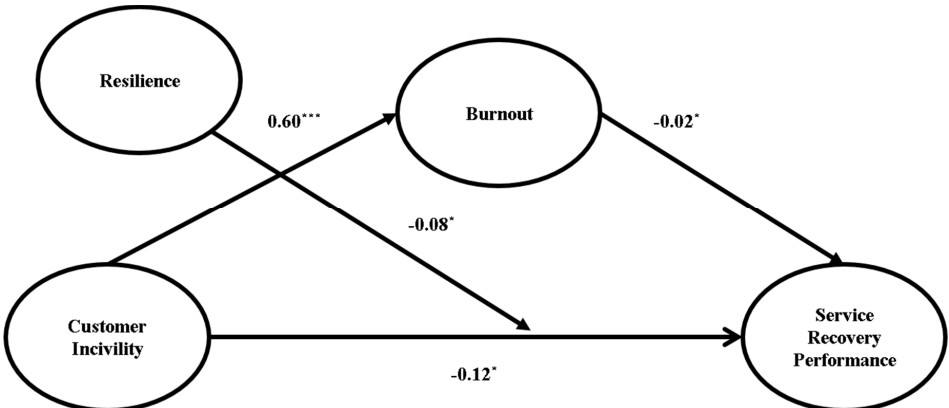

**Figure 2.** Standardized path coefficients for model with resilience as moderator of the relationship between customer incivility and service recovery performance.

**Table 5.** Standardized conditional effects for the model with resilience as moderator of the association between customer incivility and service recovery performance.

| Model: X × W→Y | Standardized Conditional Effects | | |
|---|---|---|---|
| | Estimate | S.E. | p |
| CI × Low levels of Resilience→SRP | **−0.21** | **0.07** | **0.003** |
| CI × Moderate levels of Resilience→SRP | **−0.13** | **0.05** | **0.015** |
| CI × High levels of Resilience→SRP | −0.04 | 0.06 | 0.432 |

Note. Boldfaced numbers indicate statistically significant models. X = I.V.; W = moderator; Y = D.V.; CI = Customer Incivility; SRP = Service Recovery Performance.

The validity of the hypothesized model was assessed by comparing it (i.e., in terms of BIC and AIC comparative indices) with three competing models, as described in detail in Table 6. As shown, the model with resilience was the better-fitting model compared to those which included other personal characteristics as moderators.

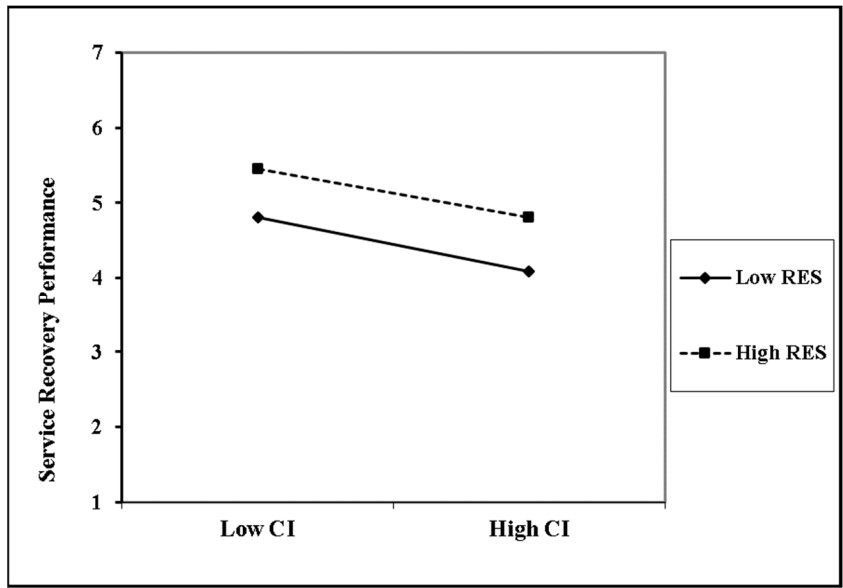

**Figure 3.** The moderating role of resilience in the relationship between customer incivility and service recovery performance.

**Table 6.** Goodness of fit indices for the selected job control moderation model and its competing models.

| Model | X × W→Y | AIC | BIC |
|:---:|:---:|:---:|:---:|
| **M1** | **CI × Resilience→SRP** | **29853.43** | **30558.37** |
| M2 | CI × NA→SRP | 29855.95 | 30558.93 |
| M3 | CI × CO→SRP | 29860.19 | 30563.12 |
| M4 | CI × PA→SRP | 29861.50 | 30564.39 |

Note. In bold the selected model; X = I.V.; W = moderator; Y = D.V.; CI = customer incivility; NA = negative affectivity; CO = customer orientation; PA = positive affectivity; BIC = Bayesian Information Criterion; AIC = Akaike Information Criterion.

## 4. Discussion

The present study analyzed whether CSSs, including CI, could influence SRP through burnout symptoms and whether two personal resources, namely CO and resilience, could help frontline employees to provide a good SRP. Several findings emerged from this research which makes a meaningful contribution to the existing body of knowledge on CSSs and SRP (see Table 7 for a summary of results).

First, differently from the other CSSs, CI exerted a direct and negative influence on SRP. A plausible explanation may be drawn on the different occurrence of customer uncivil encounters in comparison with customer verbally aggressive interactions. Indeed, since CI is likely to occur with a higher occurrence than CVA, the accumulation of uncivil acts over time may have a stronger negative impact, despite of its lower magnitude, on SRP than isolated actions of CVA. Moreover, CI may be perceived as a more severe form of CSSs when compared to DCE, ACE, and DC. Additionally, this finding is in line with the results from previous studies which found that CI was positively related to negative job outcomes among service providers, such as reduced sale job performance [128]. The theoretical contribution of this finding is to identify the differential impact of different forms of CSSs on SRP.

**Table 7.** Summary of results for each Hypothesis. DCE: disproportionate customer expectations; ACE: ambiguous customer expectations; CVA: customer verbal aggression; DC: disliked customers; CI: customer incivility; SRP: service recovery performance.

| Hypothesis | Description | Result |
|---|---|---|
| H1a. | DCE will be directly and negatively associated with SRP | Not accepted |
| H1b. | ACE will be directly and negatively associated with SRP | Not accepted |
| H1c. | DC will be directly and negatively associated with SRP | Not accepted |
| H1d. | CVA will be directly and negatively associated with SRP | Not accepted |
| H1e. | CI will be directly and negatively associated with SRP | Accepted [a] |
| H2a. | DCE will negatively influence SRP through burnout symptoms | Accepted [b] |
| H2b. | ACE will negatively influence SRP through burnout symptoms | Accepted [b] |
| H2c. | DC will negatively influence SRP through burnout symptoms | Accepted [c] |
| H2d. | CVA will negatively influence SRP through burnout symptoms | Accepted [b] |
| H2e. | CI will negatively influence SRP through burnout symptoms | Accepted [c] |
| H3. | CO will directly and positively influence SRP | Accepted [a] |
| H4. | Resilience will buffer the negative effects of CI on SRP | Accepted [c] |

Note. [a] = *** $p < 0.001$; [b] = ** $p < 0.01$; [c] = * $p < 0.05$.

Second, all CSSs increased the risk for employees of experiencing burnout symptoms and, in turn, reduced SRP. This is in accordance with the results from prior studies analyzing the associations between CSSs, burnout and SRP [6,57,70,71]. For instance, Karatepe and colleagues [57] showed that CVA intensified emotional exhaustion which, in turn, produced reduced SRP among frontline hotel employees. Similarly, Kim and co-workers [72] revealed that CSSs negatively influenced frontline employees' service recovery efforts through emotional exhaustion. Our findings can be explained in the light of the COR theory [16]. Since CSSs progressively exhausted employees' emotional and cognitive resources, individuals who became burned-out might need to recover from such stressful experiences. As a result, resource-depleted workers might be unwilling to continue depleting their resources investing their limited energies in service recovery efforts and, thus, they might reduce the quality of the SRP provided in the attempt of preserving their remaining resources. Prior investigations have concentrated on neither incivility from intra-organizational members or on task-related workplace stressors [86] or on severe forms of aggression from outsiders who have no legitimate relationships to the business, such as robbery-related violence [129,130], with less attention given to stressors from customers. By shedding light on the CSS-SRP relationship, the current research contributes to address this gap and provide empirical evidence for the COR theory regarding threatening customer encounters and their impact on individual and organizational outcomes.

Third, individual-level CO was directly and positively related to SRP. This finding is consistent with the results from a few Korean hotel investigations [15,131] which showed the presence of a positive relationship between organizational-level CO and SRP. By replicating Choi and colleagues' [56] findings, the current study is among the first investigations to reveal that individual-level CO directly fosters SRP. Indeed, CO enables workers to maintain high-quality SRP [78] by predisposing them to display customer-satisfying behaviors [94] and seek additional resources to find solutions to customers' problems [92]. This means that highly customer-oriented employees are intrinsically self-motivated to invest energies to satisfy customers' needs and expectations [81]. This study contributes to existing research by providing further support to a limited but increasing body of empirical evidence [81] which suggests the relevance of CO as a critical tool in pursuing SRP.

Fourth, resilience buffered the detrimental effects of CI on SRP. According to the COR theory [66], resilience represents a coping personal resource which may help service providers to proactively prepare themselves for—and effectively manage—challenging customer encounters by utilizing, by developing, and by maintaining their resource caravans. A further possible explanation for the positive influence of resilience is its role in employees' attribution processes. In accordance with the Cognitive Appraisal Theory [132], individuals engage automatically in primary appraisal to evaluate the significance attributed to an environmental situation by sensing whether the condition exceeds

their resources. When something in the environment is perceived as a condition significant to a person's well-being, the individual utilizes a second appraisal process to evaluate the availability of coping resources and develop reactions to the event. Using this framework, highly resilient employees may appraise CI as less threatening due to their natural disposition to have optimistic thinking and be able to regulate their emotional exhaustion [103,104,133,134]. Moreover, resilience may represent a personal coping resource in the secondary appraisal process [132] as resilient employees tend to thrive in challenging circumstances [135,136] and easily utilize available resources to restore customers' satisfaction with the service. In other words, resilient employees are likely to respond to customers' discontent with effective solutions. Additionally, employees with high resilience levels can develop effective relaxation skills to stay calm and positive, thereby making right and timely decisions to effectively deal with uncertain and problematic service failures [109]. To the best of our knowledge, the role of resilience in moderating the relationship between CI and SRP had not been yet investigated before of the present research. Nevertheless, it is important to do so, because results may offer some interesting implications for recruitment and organizational interventions aimed at preserving highly performing employees. Thereby, we extended existing incivility literature by showing that resilience can mitigate the detrimental effects of CI on employees' SRP perceptions.

Taken together these results, drawing on the COR theory [16,66] and the psychology of sustainability [17,18], we extended existing literature by empirically testing how certain personal resources relevant for customer-contact employees, namely CO and resilience, may enforce employees' SRP and protect them from the harmful impact of CI.

### 4.1. Strengths and Limitations

The current study has a number of strengths. It gives an original contribution to the existing literature on CSSs: this is one of the few studies assessing the influence that different CSSs, including CI, may have on the development of burnout symptoms and, in turn, on SRP. Furthermore, to the best of our knowledge, this is one of the first study to examine the direct influence of individual-level CO on SRP and the first research to investigate the buffering role of resilience on the CSSs-SRP relationship. Additionally, this is one of the first investigation to analyze the influence of stressors from customers on a job-related outcome within the psychology of sustainability framework [17,18].

However, our findings are also subjected to some limitations. First, the current study relied on one source of information for data gathering which might contribute to common method bias [137]. Although common method bias is seldom severe enough to compromise the validity of the results [138], we followed Podsakoff and colleagues' [139] recommendations regarding questionnaire design to decrease this bias. Future research would benefit from integrating different information sources. Second, the cross-sectional design of this investigation did not allow us to infer causal relationships. Therefore, future studies should overcome this limitation by using longitudinal techniques and assess employees' well-being and SRP before CSSs take place, in order to more thoroughly interpret how experiencing CSSs may impact on these perceptions. Third, results cannot be generalized to specific working populations since the current research was conducted on a sample of working students employed in different customer-contact areas. Thereby, replications should be carried out in specific professional contexts through comparative studies addressed at full-time employees from different organizations and national contexts. Fourth, possible selection bias due to the voluntary participation into the research cannot be ruled out. Thus, it is possible that those who experienced demanding customer encounters were more motivated to respond and, as such, were overrepresented. Fifth, since the majority of subjects were women, and gender has been found to affect the levels of burnout among service workers [71,140], this might have partially influenced our findings. However, the gender distribution in our sample is highly representative of the Italian customer service workforce. Sixth, other personal characteristics (e.g., emotional intelligence) [73] and job resources (e.g., human capital sustainability leadership, workplace relational civility) [20,141] which we did not measure could influence the associations between CSSs, burnout and SRP. We must leave to future work the task of

addressing the questions of whether other personal or job resources may influence these relationships. Moreover, since the current study is concentrated on the individual level only, future investigations should integrate different levels of analysis into a multilevel nature model.

*4.2. Practical Implications*

There are several important implications of this empirical investigation for service managers.

Firstly, since encounters with rude customers may directly affect employees' SRP, organizations should take steps to prevent CI from occurring in the first place. The management should institute a formal written zero-tolerance policy for customer mistreatment, distinguishing reasonable from unreasonable customers' demands [56]. Furthermore, managers should ensure they are providing quality customer services and they are soliciting frequent feedbacks from clients to detect service failures.

Secondly, since CO can directly influence SRP and resilience can buffer the detrimental effects of CI on SRP, interventions should concentrate on the enhancement of these individual resources to build strengths and promote well-being [73]. On the one hand, service firms can greatly benefit from selecting and hiring highly customer-oriented and resilient employees for customer-contact positions. By establishing more through instruments to evaluate applicants' CO and resilience levels and by emphasizing these personal resources as critical credentials that candidates should have, HR representatives can select the most suitable candidates and facilitate a better job-person fit. On the other hand, organizations could cultivate high CO attitudes through supervisory support and mentoring to instill higher levels of CO among low-CO workers [86]. Additionally, companies should provide their employees with psychological resilience training programs [9] and structured training sessions aimed at improving their recovery skills to successfully face customer interactions, perceive difficult complainants, respond to customers' requests and, in the meantime, foster customer-oriented behaviors. To enhance further positive individual resources, training programs should foster service providers' emotion regulation skills, coping strategies, relational management skills as well as negotiations abilities to de-escalate critical events, acknowledging the importance of preventing confrontations.

Thirdly, managers should facilitate "environmental conditions that support, foster, enrich, and protect" (i.e., caravan passageways) ([142], p. 176) service providers' resources, thereby preventing burnout symptoms due to CSSs which undermine SRP. In doing so, organizations should enable workers to rely on organizational resources to face unpleasant encounters without depleting their own emotional reserves. Thereby, interventions should be implemented at the group level to help workers build strong bonds and the social support needed to manage demanding customer interactions [17,18]. Managers could conduct sharing and debriefing sessions (also with the support of a professional psychologist, where appropriate) with service providers where workers are stimulated to openly share their emotional experiences with difficult clients as well as their experiences of success to make them aware of one's own resources useful to face new challenges [17,18]. Moreover, supervisors should support their subordinates through regular communication and mentoring sessions aimed at analyzing negative customer encounters and at finding tailored solutions to satisfy customers' expectations. Additionally, by transferring or knowing that transferring unreasonable customers to one's own supervisors is allowed, service providers can decrease their likelihood of developing burnout following CSSs and view their supervisors as more supportive. At the organizational level, attention should be paid to promote a strong service climate [143] which supports positive relationships and gives some power back to employees [17,18]. For instance, organizations may consider enabling employees to take short breaks at their discretion after handling a difficult complainant, give systematic feedback to inform the management about stressful conditions as well as use job rotations to limit their contact with the public [78] to facilitate their recovery from emotional and cognitive resources loss. In addition, by establishing a real-time and flexible service, empowered employees can exercise their discretional power to promptly respond to customer's problems with solutions, make necessary remedies to satisfy customers' requests and defend themselves against uncivil customers [144,145]. Empowerment may

also be perceived as a sign of organization's trust in and support toward its employees, thus increasing perceived organizational support and, in turn, SRP [146]. In other words, a service firm could utilize organizational flexibility and resources rather than consuming employees' resource reservoirs when CSSs are abnormally demanding to sustain healthy and, thus, more productive workers.

## 5. Conclusions

Promotion of individual well-being represents one of the 17 sustainable development objectives of the United Nations which is acknowledged as vital for facilitating world sustainable development [147]. Within the psychology of sustainability framework [17,18], building workers' strengths and, as a result, their flourishing is essential from a primary prevention perspective [73]. Given the increasing market competitiveness, companies strive to deliver high-quality service recovery performance to resolve inevitable service failures to the satisfaction of the customer. The current research provides some meaningful insights into the mediating role of burnout in the CSSs-SRP relationship. From an applicative perspective, managers should recognize the importance of supporting service providers during negative customer encounters by fostering a resource-high work setting which enables them to rely on organizational resources (e.g., supervisor support and greater latitude in the SRP delivery process) to facilitate the handling of CSSs, thereby protecting them from burnout. This may be advantageous from an organizational perspective because such employees in this workplace are likely to replenish their emotional resources more easily and, therefore, be more productive and better able to turn dissatisfied customers into satisfied ones. Additionally, this study contributes to the existing literature by identifying two crucial personal coping resources, namely CO and resilience, which may help individuals in maintaining high SRP levels, even when confronted with CSSs. In this respect, managers should establish and capitalize on a human resource management system, which includes structured procedures (e.g., the assessment of candidates' CO and resilience during the recruiting process) as well as training programs—together with experience sharing sessions and mentoring sessions—to promote the enhancement of employees' personal resources to support them in overcoming CSSs. Indeed, identifying and fostering protective personal resources is crucial to promote service providers' well-being and healthy workplaces

**Author Contributions:** Conceptualization, V.S.; Methodology, V.S., I.S. and P.A.; Data curation, V.S.; Formal analysis, V.S.; Project administration, V.S., I.S. and P.A.; Writing—original draft, V.S.; Writing—review and editing, I.S., P.A.; Visualization, V.S.; Supervision, I.S., P.A.

**Funding:** This research received no external funding.

**Conflicts of Interest:** The authors declare no conflict of interest.

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
