# Peer review of "The Role of Service Providers’ Resilience in Buffering the Negative Impact of Customer Incivility on Service Recovery Performance"

_sustainability, doi:10.3390/su11010285_

Round 1

Reviewer 1 Report

Thank you for the opportunity to review this article. I realize that a great work and time has been devoted to this paper. It has a lot of strengths, but I think that some changes should be recommended. 

First, the title does not orient the readers well about the content of the paper. Perhaps it is too long. "Investigating the..." is obvious and can be deleted for focusing on the main topic. 

The abstract is too short. You should consider that some readers only will access the abstract. Please, inform them about your findings in a more detailed manner. 

The literature review is one of the major flaws of the paper. Firstly, some of the references that you cite are too old. Even though the most relevant studies should be referenced, also the RECENT research must be included. Please, revise the current volumes from the most relevant journals and include the update literature.

Secondly, all the section and some paragraph are too long. In an example, some of them extend over one page. It could be easier for the readers is paragraphs are shortened. 

Sample: it is difficult for me to understand if the participants are students or workers. 

You should better explain for readers how participants have been recruited. 

The results section is strong and rigorous. 

In the discussion, you should comment about the potential bias due to the high percentage of women in the sample.  

Author Response

Reviewer 1

1.      First, the title does not orient the readers well about the content of the paper. Perhaps is too long, “Investigating the” is obvious and can be delated for focusing on the main topic.

We would like to thank the reviewer for his/her useful suggestion. We shortened the title as follows: “The role of service providers’ resilience in buffering the impact of customer incivility on service recovery performance.”

2.      The abstract is too short. You should consider that some readers only will access the abstract. Please, inform them about your findings in a more detailed manner.

We appreciate your constructive comment. Accordingly, we modified our abstract by explaining our findings in a more detailed manner and by including practical implications. Now, the abstract is a total of 200 words (the maximum allowed) and it reads as follows:

 â€śIn the service sector, customer-related social stressors may weaken employees’ well-being, impairing job-related outcomes. Drawing on the Conservation of Resources theory and on the psychology of sustainability, fostering personal resources become critical to flourish service providers who can effectively manage such job demands. This study investigated how customer-related social stressors and customer orientation influence service recovery performance and whether resilience buffers the negative effects of customer incivility on service recovery performance. 157 Italian customer-contact employees completed a questionnaire analyzing: customer incivility, customer-related social stressors, resilience, customer orientation, service recovery performance. Regression analyses and SEMs were conducted. Although all customer-related social stressors indirectly and negatively influenced service recovery performance by increasing burnout symptoms, customer incivility only exerted a direct and detrimental impact on service recovery performance. Customer orientation was directly and positively associated with service recovery performance. High-resilient employees were less affected by variations in service recovery performance across customer incivility levels. Within the psychology of sustainability framework, promoting resilient workplaces is crucial to foster healthy and sustainable work settings. Service organizations can greatly benefit from providing their employees with psychological resilience training programs, cultivating high customer-oriented attitudes through mentoring sessions and hiring highly customer-oriented and resilient employees for customer-contact occupations.

Keywords: customer-related social stressors, resilience, customer orientation, service recovery performance, psychology of sustainability”

3.      The literature review is one of the major flaws of the paper. Firstly, some of the references that you cite are too old. Even though the most relevant studies should be referenced, also the RECENT research must be included. Please, revise the current volumes from the most relevant journal and include the update literature.

Thank you for your constructive criticism. However, we believe that we included updated papers from the most relevant journals published within the Work and Organization Psychological field. We conducted our literature review using two different databases, namely Scopus and PsychINFO (which are the databases more generally used within the psychological research field). To check the presence of new papers, we conducted the same literature review explained below the 19th of December. To minimize selection bias, a data extraction sheet was developed and pilot-tested on ten randomly-selected included studies and then refined accordingly. Data extraction was completed independently by the first and second authors. Disagreements about keeping or dropping papers were resolved by discussion between the two review authors; if no agreement could be reach, it was planned a third author would decide. 

The first bibliographic research was carried out including the following keywords: “service recovery performance” AND “customer-related social stressors” OR “customer incivility” OR “customer verbal aggression”. We obtained 3 papers published on academic journals using PsychINFO and 2 academic articles using Scopus. In total we obtained 4 papers (1 duplicated was removed): Choi, Kim, Lee, & Lee (2014); Kim, Paek, Choi, & Lee (2012); Yoo, Kim, Terry, & Lee (2015); Karatepe, Osmans, Yorganci, & Mine (2009).

The second literature review used the following keywords: “service recovery performance” AND “burnout” or “burn-out” or “burn out” or “stress” or “occupational stress” or “emotional exhaustion” or “cynicism”. We obtained 8 papers published on academic journals using PsychINFO and 3 academic articles using Scopus. In total, we included two new papers (two were duplications), namely: Karatepe, O. M.; Yorganci, I.; Haktanir (2009); Kim, T.; Jung-Eun Yoo, J.; Lee, G.; Kim (2012).

The third bibliographic research considered the following keywords: “service recovery performance” AND “resilience” or “customer orientation”. We obtained 7 academic papers using PsychINFO and 16 document results using Scopus, 4 were duplications of previously found articles. We excluded the papers which were focused on general customer service orientation because our study was specifically focused on individual-level customer orientation. We excluded papers analyzing different personal resources, such as job resourcefulness. 4 papers were suitable for our research focus, namely: Kim, T.; Karatepe, O. M.; Lee, G.; Lee, S.; Hur, K.; Xijing (2017); Kim, Paek, Choi, & Lee (2012); Hammami, & Triki (2011); Cheng, Cheng, Chang, 2008; Yavas, U.; Babakus, E.; Karatepe, 2010;

Moreover, we based our work on a systematic review entitled “The impact of customer incivility and verbal aggression on service providers: A systematic review.”, which is in press on Work- A journal of prevention, assessment and rehabilitation.

Nevertheless, we acknowledge that some articles which were dated could be substituted with more recent papers. Therefore, we made different changes to address your concern.

We changed Tax et al. (1998) with Tang, Chang, Huang, & Zhang (2018)

We substituted Andreassen (1999) with Valenzuela, Vaswuez-Parraga, Llanos, & Vilches (2006)

We changed Benoy (1996) with Nair (2015)

We substituted Hobfoll & Shirom (1993) with Yoo et al. (2015)

We changed Hart et al. (1990) with Wang, Hsu, & Chih (2014)

We added Rod & Ashill (2010) and the following sentence (p.3) : “Additionally, service providers can learn from recovery services and improve their performance – in terms of recovery speed and recovery quality -, accordingly [11].”

Considering the relevance of these papers, we decided to keep the following dated sources: Hobfoll, 1989; Watson, 1988; Lee, & Ashforth, 1996; Wright, & Cropanzano, 1998

4.      Secondly, all the sections and some paragraphs are too long. In an example, some of them extend over one page. It could be easier for readers is paragraphs are shortened.

We would like to thank you the reviewer for his/her useful comment. We shortened all paragraphs in order to reduce each of them to maximum one page. To facilitate reading, we created sub-paragraphs.  Furthermore, for the ease of the reader, we included a paragraph presenting the organization of the paper, as follows (p.3): “The rest of this paper is organized as following: (1) the next section, briefly reviews the related literature around the relationships between CCSs and SRP as well as the protective role of resilience and CO in maintaining SRP, and then develops direct, mediating and moderating hypotheses. (2) The second section describes the empirical setting of this study, including materials, methods. (3) The third section presents the statistical analyses and reports the empirical results, (3) The final section discusses findings, limitations and practical implications, and then concludes the study.”

The revised version of our manuscript includes the following sub-paragraphs: 1.1. Service recovery performance (see pp. 3-4); 1.2. The detrimental impact of customer-related social stressors on service recovery performance (see pp.4-5); 1.3. The mediating role of burnout (see pp. 5-6); 1.4. The importance of personal resources (p.6); 1.5. The positive influence of customer orientation (pp. 6-7); 1.6. Resilience as a moderator (see pp. 7-8)

5.      Sample: it is difficult for me to understand if the participants are students or workers. You should better explain for readers how participants how been recruited

Thank you for your valuable criticism. Research participants were working students who were working in a retail sale or restaurant service occupation for at least 6 months. We rewrote the sample and procedure section to provide more information about research participant recruitment, as follows (see p. 9): “Participants were psychology students who were recruited using academic newsletter and e-mail system or were enrolled in psychology courses at University of Pavia. To participate, students were required to be working in a retail sale (e.g., shop assistant, cashier) or restaurant service (e.g., waiter, bartender) job for at least 6 months, be 18 years of age or older, and have at least a moderate amount of contact with the public, so that certain stressful events were likely to occur. Working students received extra course credit for taking part in the research. Once they voluntarily agreed to participate, we obtained informed consent from them and ensured them the anonymity and confidentiality of their responses. Then, they were invited to complete questionnaires which were administrated by professional trainees in Psychology within a laboratory setting. In total, 157 Italian students employed in customer-contact jobs took part to the present study. We excluded 3 participants who did not meet the study criteria (e.g., not employed in customer services) and 2 participants because they did not complete at least the 60% of the survey. The majority of respondents were female (78.30%) who spent on average the 85.91% of their working time in direct contact with the public. They were mainly employed as waiters or shop assistants. The average age was 25.27 years (SD=5.59) with an average job tenure of 4.53 years (SD=4.99) and an average job tenure in the current position of 2.95 years (SD=4.99). “

6.      The results section is strong and rigorous.

Thank you very much for your positive feedback. To better explain our statistical analyses, we introduced a new paragraph entitled “Statistical analyses”. Now the revised paper reads as follows (p. 13): “The data were first explored for descriptive statistics and correlations using SPSS 20 statistical program for Windows [122]. Then, all CSSs were added to a regression model to simultaneously predict SRP. To test the mediating role of burnout in the relationship between CSSs and SRP, we conducted mediation models using Mplus Version 7 [123]. Fit models were examined using the root mean squared error of approximation (RMSEA, 124; values of .05 are taken as good fit, .05-.08 as moderate fit, 125), the standardized root mean square residual (SRMR; a value less than .08 is considered a good fit, 125), the comparative fit index (CFI, 126; values between .90 and .95 indicate acceptable fit, 125) and the Tucker-Lewis index (TLI, values between .90 and .95 indicate acceptable fit, 127). To test direct effects of personal characteristics (i.e., CO, resilience, PA and NA) on SRP in a combined model, all individual features were added to a regression model having SRP as dependent variable. The regression analyses were conducted using enter variable selection in which variables were randomly selected and entered since we did not have research evidence to hypothesize a certain order. Subsequently, to examine the moderating effect of resilience on the relationship between CI and SRP, a moderation model was carried out using Mplus Version 7 [123], while controlling for CO, PA and NA. The goodness of the model was evaluated by comparing it in terms of BIC (Bayesian Information Criterion) and AIC (Akaike Information Criterion) comparative indices with three competing models. Lower values of AIC and BIC indicate a better fit and the model with the lowest AIC and BIC is the best fitting model.”

7.      In the discussion, you should comment about the potential bias due to the high percentage of women in the sample.

We appreciate this useful comment and we recognize that the high percentage of women in our sample is a limitation of our study. Indeed, we could not include gender as a control variable because nearly all participants were female. Accordingly, we added the following limitation to the discussion (p.20):

 â€śFifth, since the majority of subjects were women, and gender has been found to affect the levels of burnout among service workers [71,140], this might have partially influenced our findings. However, the gender distribution in our sample is highly representative of the Italian customer service workforce.”

Reviewer 2 Report

The paper is a good contribution to the literature and relevant conclusions are achieved. The authors investigated whether resilience may buffer the negative effects of customer incivility on service recovery performance. The paper has potential, however some comments must be taken into account before being considered for publication:

·         I would recommend to present the abstract in a single paragraph;

·         Reduce the number of keywords;

·         Introduction must be improved. The introduction is too long. Do not use subchapters, for example a new chapter for literature review;

·         Parts of text of the introduction must be moved to other chapter;

·         In the introduction, include a paragraph presenting the organization of the paper;

·         In the introduction, the authors could include a definition for sustainability (see CRUZ and MARQUES, 2014);

·         All abbreviations must be presented in the text;

·         The novelty should also be highlighted in the text;

·         Explain better the figure 1;

·         More information about the case study must be included in the text (e.g. GDP, …);

·         Justify better the methodology adopted and limitations;

·         The authors must explain better the statistical analysis, including the sign of the variables;

·         In the end include a table summarising the results for each hypothesis;

·         More recommendations for decision-makers were expected in the conclusions;

·         The references must be improved and homogenised and in line with author guidelines. For example, some issues are missing.

References:

CRUZ, N.; MARQUES, R. (2014). Scorecards for sustainable local governments. Cities. Elsevier. ISSN: 0264-2751. Vol. 39, pp. 165–170.

Author Response

Reviewer 2

1.      I would recommend to present the abstract in a single paragraph.

We appreciate your constructive comment. Based on your recommendation, we presented our abstract in a single paragraph. Additionally, we modified our abstract by explaining our findings in a more detailed manner and by including practical implications. Now, the abstract is a total of 200 words (the maximum allowed) and it reads as follows:

 â€śIn the service sector, customer-related social stressors may weaken employees’ well-being, impairing job-related outcomes. Drawing on the Conservation of Resources theory and on the psychology of sustainability, fostering personal resources become critical to flourish service providers who can effectively manage such job demands. This study investigated how customer-related social stressors and customer orientation influence service recovery performance and whether resilience buffers the negative effects of customer incivility on service recovery performance. 157 Italian customer-contact employees completed a questionnaire analyzing: customer incivility, customer-related social stressors, resilience, customer orientation, service recovery performance. Regression analyses and SEMs were conducted. Although all customer-related social stressors indirectly and negatively influenced service recovery performance by increasing burnout symptoms, customer incivility only exerted a direct and detrimental impact on service recovery performance. Customer orientation was directly and positively associated with service recovery performance. High-resilient employees were less affected by variations in service recovery performance across customer incivility levels. Within the psychology of sustainability framework, promoting resilient workplaces is crucial to foster healthy and sustainable work settings. Service organizations can greatly benefit from providing their employees with psychological resilience training programs, cultivating high customer-oriented attitudes through mentoring sessions and hiring highly customer-oriented and resilient employees for customer-contact occupations.”

2.      Reduce the number of keywords

Thank you for this comment. Instructions for authors indicate to include three to ten pertinent keywords after the abstract. However, based on your suggestion, we deleted the keyword “customer incivility”, leaving the wider term “customer-related social stressors”. We included “resilience” and “customer orientation” because they are the investigated personal resources, whereas “service recovery performance” represents our outcome. Moreover, we deleted the keyword “healthy organizations”, leaving “psychology of sustainability” that is one of the keywords suggested for this special issue. The revised version of our manuscript has the following keywords: “customer-related social stressors, resilience, customer orientation, service recovery performance, psychology of sustainability”

3.      Introduction must be improved. The introduction is too long. Do not use subchapters, for example chapter for literature review. Parts of the text in the introduction must be moved to other chapter

We would like to thank the reviewer for his/her useful comment. We shortened all paragraphs in order to reduce each of them to maximum one page. We reduced the length of Introduction to one page, moving parts of the text in the introduction to other sub-chapters. Additionally, to facilitate reading, we created new sub-paragraphs.  The revised version of our manuscript includes the following sub-paragraphs: 1.1. Service recovery performance (see pp. 3-4); 1.2. The detrimental impact of customer-related social stressors on service recovery performance (see pp.4-5); 1.3. The mediating role of burnout (see pp. 5-6); 1.4. The importance of personal resources (p.6); 1.5. The positive influence of customer orientation (pp. 6-7); 1.6. Resilience as a moderator (see pp. 7-8)

In the introduction, include a paragraph presenting the organization of the paper

Thank you for your valuable suggestion. Accordingly, we added the following paragraph to the introduction (p. 3): “The rest of this paper is organized as following: (1) the next section, briefly reviews the related literature around the relationships between CCSs and SRP as well as the protective role of resilience and CO in maintaining SRP, and then develops direct, mediating and moderating hypotheses. (2) The second section describes the empirical setting of this study, including materials, methods. (3) The third section presents the statistical analyses and reports the empirical results, (3) The final section discusses findings, limitations and practical implications, and then concludes the study.”

4.      In the introduction, the authors could include a definition for sustainability (see CRUZ and MARQUES, 2014)

Thank you for your comment. We read the paper you suggested to us and we included a note containing the definition for sustainability to clarify the meaning of this term form a psychological perspective, as follows (pp.): “From a psychological perspective, the word “sustainability” refers not only to balance current objectives with future aims without jeopardizing the latter by avoiding harmful actions within the ecological and socio-economic environment [18], but also to promote individual well-being by stimulating their enrichment, growth, and flexible change and by facilitating the acquisition of resources [17].”

5.      All the abbreviations must be presented in the text

Thank you for this useful comment. We revised the whole manuscript in order to define in parentheses all abbreviations the first time they appeared in the text and we used consistently thereafter. For instance, please see the following (p. 4): “(1) disproportionate customer expectations (DCE; i.e., customers’ behaviors challenging what is considered reasonable from workers’ perspectives); (2) ambiguous customer expectations (ACE; i.e., unclear customers’ requests); (3) disliked customers (DC; aversions customer-contact employees have to unpleasant customers who cause interruptions); (4) customer verbal aggression (CVA; i.e., verbal abuse perpetrated by a customer, with the clear intent to hurt a worker through offensive verbal expressions).“ Moreover, in order to enable the reader to understand the content of Tables and Figures we wrote more detailed Notes. For example, we modified the Note of the first Table, as follows (p. 40): “Note. Boldfaced numbers on the diagonal represent Cronbach’s alpha; M= means; SD= standard deviation; *p<05; **p<.01; ***p<.001. CI= customer orientation; CVA= customer verbal aggression; DCA= disproportionate customer expectations; ACE= ambiguous customer expectations; DC= disliked customers; SRP= service recovery performance; CO= customer orientation; PA= positive affectivity; NA= negative affectivity.”

6.      The novelty should also be highlighted in the text

According to your suggestion, we modified the Introduction, underling novelty at the beginning, as follows (pp. 2-3): “To date, the majority of research has focused on customer perceptions [11], whereas only a few studies have analyzed this topic from the service provider perspective [12].  To date, only a few studies have examined the direct association between CO and SRP, with previous investigations on this topic predominantly concentrating on organizational-level CO [13-15]. Moreover, to the best of our knowledge, no previous studies have analyzed resilience as a possible buffer for negative effects of customer incivility (CI) on SRP. To fil these gaps, drawing on the Conservation of Resource theory (COR) [16] and adopting the service provider perspective, the current study investigated how CSSs and individual-level CO influence SRP and whether resilience buffer the detrimental influence of CI on employees’ SRP. Furthermore, this is one of the first studies to analyze this topic within the psychology of sustainability[1] framework [17, 19-20] which represents a promising research area for promoting healthy organizations [18,19] and improving employees’ quality of life [21], all factors that are conducive to successful business [22,23].”

We underlined the contribution provided also in the sub-paragraphs, as follows (p. 4): “To date, there are various calls for more research regarding the factors stimulating employees’ SRP [40,41]. By providing empirical evidence for the positive influence of CO and resilience on SRP, our study’s analysis of SRP makes a meaningful contribution. (…)  In doing so, the aim of the current study was to extend the original model through the inclusion of CI.” See also (pp. 5-6): “To date, only a moderate amount of empirical research has examined the associations between CSSs, staff burnout, and SRP [6,70,71]; with a paucity of studies analyzing these variables in a single framework and supporting the role of burnout symptoms in mediating the impact of different CSSs on SRP [57,72]. To fil this gap, one of the main purposes of the current study was to investigate whether different CSSs would lead employees to experience burnout which, in turn, would decrease their ability to provide good SRP. “or (p. 7) “However, to date, to the best of our knowledge, the only study investigating the direct influence of individual-level CO on SRP was conducted by Choi and colleagues [56], obtaining the same result. This suggests that further empirical investigation on the direct relationship between individual-level CO and SRP is required.”

We underlined the contribution of our study also in the discussion section. For instance, on pages 16-18, you read: “The theoretical contribution of this finding is to identify the differential impact of different forms of CSSs on SRP.” (..) “By replicating Choi and colleagues’ [56] findings, the current study is among the first to reveal that individual-level CO directly fosters SRP. This study contributes to existing research by providing further support to a limited but increasing body of empirical [81] which suggests the relevance of CO as a critical tool in pursuing SRP. (…) Prior investigations have concentrated on neither incivility from intra-organizational members or on task-related workplace stressors [86] or on severe forms of aggression from outsiders who have no legitimate relationships to the business, such as robbery-related violence [129,130], with less attention given to stressors from customers. By shedding light on the CSS-SRP relationship, the current research contributes to address this gap and to provide empirical evidence for the COR theory regarding threatening customer encounters and their impact on individual and organizational outcomes. (…) To the best of our knowledge, the role of resilience in moderating the relationship between CI and SRP had not been yet investigated before of the present research. Nevertheless, it is important to do so, because results may offer some interesting implications for recruitment and organizational interventions aimed at preserving highly performing employees. Thereby, we extended existing incivility literature by showing that resilience can mitigate the detrimental effects of CI on employees’ SRP perceptions.  (..) Taken together these results, drawing on the COR theory [16,66] and on the psychology of sustainability [17,19], we extended existing literature by empirically testing how certain personal resources relevant for customer-contact employees, namely CO and resilience, may enforce employees’ SRP and protect them from the harmful impact of CI. “

The novelty is underlined also in the strengths section (p. 19): “The current study has a number of strengths. It gives an original contribution to the existing literature on CSSs: this is one of the few studies assessing the influence that different CSSs, including CI, may have on the development of burnout symptoms and, in turn, on SRP within the psychology of sustainability framework [17,19]. Furthermore, to the best of our knowledge, this is one of the first study to examine the direct influence of individual-level CO on SRP and the first study to investigate the buffering role of resilience on the CSSs-SRP relationship. Additionally, this is one of the first investigation to analyze the influence of stressors from customers on a job-related outcome within the psychology of sustainability framework [17,19].”

7.      Explain better the Figure 1

We would like to thank you for your constructive comment. Accordingly, we modified Figure 1, including the sign of the variables and we changed its Note to explain better the content of the Figure in the text, as following (p. 8): “As a conceptual framework, Figure 1 illustrates our proposed model, incorporating our hypothesized relationships. We expected that DCE, ACE, DC, CVA and CI would directly (H1a, H1b, H1c, H1d, H1e, respectively) and indirectly (through burnout symptoms; H2a, H2b, H2c, H2d, H2e, respectively) negatively influence SRP. Moreover, we expected that CO would be directly and positively associated with SRP (H3). Additionally, we hypothesized that resilience would moderate the relationship between CI and SRP (H4).”

8.      More information about the case study must be included in the text (e.g., GDP,...)

Thank you for your comment. According to your recommendation, we added the following in the Introduction section (p. 2): “In this scenario, service recovery performance (SRP) plays a crucial role in recovering customers' loyalty and satisfaction [3], especially among Western countries where the service sector represents the main employment area, accounting for more than 60% of global Gross Domestic Product (GDP; 4). For instance, in Italy such sector contributed around 70% to the employment rates and around 66% to the GDP in 2017 [5].”

Furthermore, we reworded the sample description section in order to provide more detailed information about our research participants, as following (p. 9): “Participants were psychology students who were recruited using academic newsletter and e-mail system or were enrolled in psychology courses at University of Pavia. To participate, students were required to be working in a retail sale (e.g., shop assistant, cashier) or restaurant service (e.g., waiter, bartender) job for at least 6 months, be 18 years of age or older, and have at least a moderate amount of contact with the public, so that certain stressful events were likely to occur. Working students received extra course credit for taking part in the research. Once they voluntarily agreed to participate, we obtained informed consent from them and ensured them the anonymity and confidentiality of their responses. Then, they were invited to complete questionnaires which were administrated by professional trainees in Psychology within a laboratory setting.”

9.      Justify better the methodology adopted and limitations

Thank you for your valuable comment. We justified better our methodology by explaining the reasons why we adopted these scales. Now, you read as follows (pp. 9-10): “All the variables were measured using scales taken or adapted from previously validated and published instruments. To assess resilience, trait affectivity, CSSs and burnout, we used the Italian validated version of such scales. Because the other scales were originally written in English, they were subjected to a back and forward translation process. Firstly, the original English items were translated in Italian. Then, the forward translation was reviewed by a bilingual (in English and Italian) expert panel. Items that were suspected to be particularly sensitive to translation issues across cultures were translated back to English by an independent translator. The resulting version of the questionnaire was administrated to 30 pre-test respondents who were students employed in customer-contact occupations and, therefore, representative of research participants. They were systematically debriefed by asking them – for each item- what they thought was the meaning of a certain item, whether they could re-word that item using their own words, what sprang to their mind when they heard a specific expression, how they selected their answer. Finally, the modified version of the survey was discussed through two focus groups conducted by an experienced psychologist.”

The Connor-Davidson Resilience scale has been validated in Italian samples, so we added the following (p.10): “Resilience was assessed through 10 items from the Italian version [110] of the Connor-Davidson Resilience Scale [111].” Similarly, regarding the PANAS we added the following (pp. 10-11): “the Italian version [112] of the Positive and Negative Affect Scale [PANAS, 113]. This measure has been validated in Italian samples [112].” Regarding the scale assessing burnout symptoms, we added the following (p.12): “Burnout was measured using ten items from the Italian version [119] of the Maslach Burnout Inventory [120]”

Regarding the customer orientation scale, we added the following (p.10): “This scale has been broadly utilized by previous psychological studies, showing a satisfactory internal consistency [e.g., 56,77,78].”.

To justify better the adoption of Incivility from Customer Scale, we added the following sentences to the scale description (see p. 13): “This measure was developed based on a sample of working students who met criteria for research recruitment very similar to those applied to select our participants (i.e., working in retail sale or restaurant service occupations for at least 6 months). In addition, it has been found to have good reliability [e.g., 116] and had a very good reliability in the present study (α =.92).”

To justify the choice of using the Customer-related social stressors scale, we added the following (see pp. 13-14): “Customer-related social stressors were assessed through 22 items from the Italian version [117] of the Customer-related social stressors scale [CSSs scale, 6]  (..) We utilized this scale because our study was theoretically drawn on Dormann and Zapf’s [6] classification that we decided to adopt since it reflects general categories suitable for different work environments and covers a broad range of customer behaviors. Additionally, this measure was developed based on a sample of three service sectors’ employees (i.e., travel agency employees, shoe store sales clerks, and flight attendants) and, therefore, it was applicable for our sample. This scale has been widely used by previous studies on CSSs, showing a high degree of internal consistency among the construct items [e.g., 56,78,118]. In the current study, the reliability of the whole scale was .92.”

Regarding the service recovery performance scale, we added the following (p. 12): “This scale, which has the advantage of its short length (5 items), has been widely applied by previous psychological studies which assessed SRP within service organizations [e.g., 56,59,77,78].”.

We added the following limitations to the discussion (p. 20): “Fifth, since the majority of subjects were women, and gender has been found to affect the levels of burnout among service workers [71,140], this might have partially influenced our findings. However, the gender distribution in our sample is highly representative of the Italian customer service workforce (…) Moreover, since the current study is concentrated on the individual level only, future investigations should integrate different levels of analysis into a multilevel nature model.”

We justified better one important limitation, as following (p. 19): “First, the current study relied on one source of information for data gathering which might contribute to common method bias [137]. Although common method bias is seldom severe enough to compromise the validity of the results [138], we followed Podsakoff and colleagues’ [139] recommendations regarding questionnaire design to decrease this bias.  Future research would benefit from integrating different information sources. Second, the cross-sectional design of this investigation did not allow us to infer causal relationships. Therefore, future studies should overcome this limitation by using longitudinal techniques and assess employees’ well-being and SRP before CSSs take place, in order to more thoroughly interpret how experiencing CSSs may impact on these perceptions.”

The authors must explain better the statistical analysis including the sign of the variables

Thank you very much for your comment. We recognize the need to better explain our statistical analyses. Our explanation reads as follows in our revised paper (sub-paragraph entitled “Statistical analyses”; p. 13): “The data were first explored for descriptive statistics and correlations using SPSS 20 statistical program for Windows [122]. Then, all CSSs were added to a regression model to simultaneously predict SRP. To test the mediating role of burnout in the relationship between CSSs and SRP, we conducted mediation models using Mplus Version 7 [123]. Fit models were examined using the root mean squared error of approximation (RMSEA, 124; values of .05 are taken as good fit, .05-.08 as moderate fit, 125), the standardized root mean square residual (SRMR; a value less than .08 is considered a good fit, 125), the comparative fit index (CFI, 126; values between .90 and .95 indicate acceptable fit, 125) and the Tucker-Lewis index (TLI, values between .90 and .95 indicate acceptable fit, 127). To test direct effects of personal characteristics (i.e., CO, resilience, PA and NA) on SRP in a combined model, all individual features were added to a regression model having SRP as dependent variable. The regression analyses were conducted using enter variable selection in which variables were randomly selected and entered since we did not have research evidence to hypothesize a certain order. Subsequently, to examine the moderating effect of resilience on the relationship between CI and SRP, a moderation model was carried out using Mplus Version 7 [123], while controlling for CO, PA and NA. The goodness of the model was evaluated by comparing it in terms of BIC (Bayesian Information Criterion) and AIC (Akaike Information Criterion) comparative indices with three competing models. Lower values of AIC and BIC indicate a better fit and the model with the lowest AIC and BIC is the best fitting model.”

Furthermore, to address the concern about the explanation of the signs of the variables, we have added the description of results obtained from correlations, as follows (see pp. 13-14): “ As shown by Table 1, all CSSs were significantly and positively correlated with each other, except for DC. Similarly, all CSS were significantly and positively associated with burnout, except for DC. This suggests that working-students who were more frequently exposed to ACE were more likely to experience also DCE, CI and CVA. Additionally, employees who were affected from ACE (r=.37, p<.01), DCE (r=.37, p<.01), CI (r=.38, p<.01) and CVA (r=.20, p<.05) might be at increased risk of developing burnout symptoms. Although SRP was negatively associated with all CSSs, the correlations with CI (r=-.29, p<.01) and DCE (r=-.17, p<.05) only were significant. This means that service providers who were confronted with CI and DCE were more likely to react by reducing their SRP. Moreover, SRP was positively related to burnout (r=-.32, p<.01), such that increased burnout symptoms resulted in decreased SRP. Regarding individual characteristics, SRP was positively associated with resilience (r=.31, p<.01), CO (r=.55, p<.01) and PA (r=.30, p<.01), whereas was negatively related to NA (r=-.19, p<.05). Additionally, burnout was positively correlated with NA (r=.34, p<.05) and negatively associated with CO (r=-.26, p<.01); PA (r=-.50, p<.05) and resilience (r=-.14, ns), although this latter personal characteristic showed a non-significant correlation with SRP. Taken together these results, it seems that service providers high in resilience, CO and PA are more likely to maintain high SRP and psychological well-being levels, even when exposed to CSSs, differently from those high in NA.

Moreover, we added the following sentence (p. 15): “CO was the only significant predictor of SRP, in a direction in line with what expected. “Additionally, to better explain the sign of the interaction term, we added the following paragraph (p.15): “Hypothesis 4 was supported: resilience buffered the relationship between CI and SRP (see Figure 2 and Table 5), as indicated by the significant interaction term (β =-0.08, p<.05) which had a negative sign and, then, indicated that resilience could exacerbated the negative effect of CI on SPR. More specifically, the conditional effects indicated that working students who reported low (β =-0.23, p<.01) or moderate (β =-0.14, p<.01) resilience levels were at higher risk of experiencing impaired SRP as a result of CI when compared with those who were higher in this dimension”.

Furthermore, we included the expected signs of variables in Figure 1.

10.  In the end include a table summarizing the results for each hypothesis

We appreciate your suggestion. Accordingly, we included Table 7 which summarizes the results for each hypothesis, as follows (p. 46):

Table 7. Summary   of results for each Hypothesis

Hypothesis

Description

Result

H1a.

DCE will be directly and negatively associated with   SRP

Not accepted

H1b.

ACE   will be directly and negatively associated with SRP

Not accepted

H1c.

DC   will be directly and negatively associated with SRP

Not accepted

H1d.

CVA   will be directly and negatively associated with SRP

Not accepted

H1e.

CI   will be directly and negatively associated with SRP

Accepted a

H2a.

DCE   will negatively influence SRP through burnout symptoms

Accepted b

H2b.

ACE   will negatively influence SRP through burnout symptoms

Accepted b

H2c.

DC   will negatively influence SRP through burnout symptoms

Accepted c

H2d.

CVA   will negatively influence SRP through burnout symptoms

Accepted b

H2e.

CI   will negatively influence SRP through burnout symptoms

Accepted c

H3.

CO   will directly and positively influence SRP

Accepted a

H4.

Resilience   will buffer the negative effects of CI on SRP

Accepted c

Note. a = ***p<.001; b=**p<.01; c=*p<.05.

11.  More recommendations for decision-makers were expected in the conclusion

Thank you for your useful comment. We had included several recommendations for service organizations within the practical implication section. However, we acknowledge that the conclusion was lacking because it did not include such recommendations. As a result, we added the following (pp. 22-23): “ The current research provides some meaningful insights into the mediating role of burnout in the CSSs-SRP relationship. From an applicative perspective, managers should recognize the importance of supporting service providers during negative customer encounters by fostering a resource-high work setting which enables them to rely on organizational resources (e.g., supervisor support and greater latitude in the SRP delivery process) to facilitate the handling of CSSs, thereby protecting them from burnout. This may be advantageous from an organizational perspective because such employees in this workplace are likely to replenish their emotional resources more easily and, therefore, be more productive and better able to turn dissatisfied customers into satisfied ones. Additionally, this study contributes to the existing literature by identifying two crucial personal coping resources, namely CO and resilience, which may help individuals in maintaining high SRP levels, even when confronted with CSSs. In this respect, managers should establish and capitalize on a human resource management system, which includes structured procedures (e.g., the assessment of candidates’ CO and resilience during the recruiting process) as well as training programs -  together with experience sharing sessions and mentoring sessions -  to promote the enhancement of employees’ personal resources to support them in overcoming CSSs. Indeed, identifying and fostering protective personal resources is crucial to promote service providers’ well-being and healthy workplaces.”

12.  The references must be improved and homogenized in line with author guidelines. For example, some issues are missing

Thank you for your useful comment. We checked all references included, following author guidelines. According to such guidelines, we do not have to include issues, but volumes only. On the instruction for authors section (available from the Journal website) you can read: References should be described as follows, depending on the type of work: Journal Articles: 1. Author 1, A.B.; Author 2, C.D. Title of the article. Abbreviated Journal Name YearVolume, page range. Available online: URL (accessed on Day Month Year).”

Round 2

Reviewer 2 Report

The authors met my expectations with their papers' revision. So I am able to recommend the publication.